# Soil Management Strategies in Organic Almond Orchards: Implications for Soil Rehabilitation and Nut Quality

Belén Cárceles Rodríguez [1],*, Víctor Hugo Durán Zuazo [1], Juan Francisco Herencia Galán [2], Leontina Lipan [3], Miguel Soriano [4], Francisca Hernández [3], Esther Sendra [3], Ángel Antonio Carbonell-Barrachina [3], Baltasar Gálvez Ruiz [1] and Iván Francisco García-Tejero [2]

1   IFAPA Centro "Camino de Purchil", Camino de Purchil s/n, 18004 Granada, Spain
2   IFAPA Centro "Las Torres", Carretera Sevilla-Alcalá del Río km 12.2, Alcalá del Río, 41200 Sevilla, Spain
3   Research Group "Food Quality and Safety", Centro de Investigación e Innovación Agroalimentaria y Agroambiental (CIAGRO-UMH), Miguel Hernández University, Carretera de Beniel, km 3.2, Orihuela, 03312 Alicante, Spain
4   Centre for Intensive Mediterranean Agrosystems and Agri-Food Biotechnology (CIAIMBITAL), Agronomy Department, University of Almería, Carretera Sacramento s/n, La Cañada de San Urbano, 04120 Almería, Spain
*   Correspondence: belen.carceles@juntadeandalucia.es; Tel.: +34-958-895-200

**Abstract:** The implementation of soil conservation measures is essential to promote sustainable crop production in the Mediterranean region. In an organic rainfed almond orchard located in Lanjarón (SE, Spain), a study carried out during 2016–2021 analyzed the influence of different soil management strategies (SMSs) (TT, traditional tillage; NT, no tillage; VF, cover of *Vicia faba*; VS, cover of *Vicia sativa*; VS-VE, cover of *Vicia sativa* and *Vicia ervilia*) on some selected physical (bulk density, available water content, and aggregate stability), chemical (pH, electrical conductivity, soil-organic content, N, P, K, and micronutrients), and biological (microbial activity) soil properties, relevant to soil health, and their implications for yield and almond quality (physical and chemical). Our results showed that the SMS with legume cover improves soil properties, which had a favorable effect on soil health. The mean almond yield was not significantly affected by the SMS applied, being 315.9, 256.4, 229.1, 212.5, and 176.6 kg ha$^{-1}$ year$^{-1}$ for TT, VF, VS-VE, VS, and NT, respectively. Regarding the almond nut quality, the strategy based on implementation of legume cover increased the almond antioxidant activity and the total polyphenol content, which would improve their nutritional value. Here we showed how the use of sustainable SMSs improved the soil properties compared to traditional tillage in rainfed organic almonds, allowing the long-term sustainability of agroecosystems while at the same time obtaining higher nutritional quality almonds.

**Keywords:** almond cultivation; conservation agriculture; legume cover crops; soil health; almond quality

## 1. Introduction

Conventional practices and intensification in the agricultural sector have been recognized as main drivers of land degradation [1,2]. This phenomenon can be defined as a decrease in the quality of the land due to anthropogenic causes and that has negative effects on agricultural productivity, the environment, human well-being, and food security [3], land degradation being a global process that extends worldwide. According to Le et al. [4], about 29% of the global land area suffers from severe land degradation, as does 25% of cropland. In the European Union (EU), unsustainable management practices have caused the degradation of between 60 and 70% of soils, reducing their potential for the provision of services [5].

In this context, soil degradation and erosion are some of the main environmental problems in Mediterranean agroecosystems, affecting their sustainability and the provision of ecosystem services supplied by them [6,7]. It can be stated that soil conservation is one

of the main environmental challenges in Mediterranean semi-arid agrosystems [8]. The characteristics of the Mediterranean climate with scarce rainfall of irregular distribution, severe droughts, and high intensity storms, together with poor soil structure on steep slopes, create favorable conditions for the development of water erosion [9–11]. This problem is further aggravated by inappropriate practices such as intensive tillage, deforestation, and overgrazing.

On the other hand, forecasts for this region regarding climate change foresee a rise in temperatures, a significant reduction in net precipitation, more variability in the distribution of rain, more severe and frequent drought events, and an increase in the frequency of extreme events in the coming years [12,13], which can significantly affect crop production [14,15]. In this sense, Ray et al. [16] estimated that 60% of global land productivity is adversely affected by climate change. For the EU, the loss of agricultural productivity due to soil erosion has been calculated at 0.43% per year, which represents an economic loss of approximately EUR 1.25 billion year$^{-1}$ [17]. Therefore, in the context of climate change, the implementation of soil conservation measures is essential to promote sustainable crop production [18,19].

Rainfed woody fruit orchards (olive, almond, and vineyards) cover a high percentage of the agricultural area in the Mediterranean region. That is, in semi-arid Mediterranean conditions, such as those of southern Spain, characterized by mild rainy winters and very hot dry summers, fruit orchards are cultivated in mono-cropping systems with a low density of plantation, the limitation of tree size by pruning, and frequent tillage to avoid transpiration losses by weeds and to promote infiltration [20–22]. Moreover, these crops are usually on steep slopes and marginal lands with shallow soils low in organic matter and limited water availability [23]. These features explain why rainfed tree orchards are related to severe soil erosion in the Mediterranean region [24]. In this sense, Maetens et al. [25] stated that fruit tree crops with conventional management is the land use with the highest runoff coefficient (5–10%) and soil losses (10–20 Mg ha$^{-1}$ year$^{-1}$). Numerous studies have documented significant rates of runoff and soil loss in tree crops in the Mediterranean region [26–28]. A 12% decrease between 1982 and 2010 in the productivity of European fruit crops has been estimated as a consequence of soil degradation [29].

According to FAOSTAT [30] the total world area dedicated to almond cultivation was 2,162,263 ha, with Spain being the country with the largest area, namely, 718,540 ha, followed by the USA with 505,858 ha. In Spain, almond is mainly concentrated in the Mediterranean coast, and about 85% is under rainfed conditions [31]. Most of these orchards are monocultures, frequently tilled, with low inputs of fertilizers, and they have low yields (about 350–400 kg of almond kernel per hectare) [32].

On the other hand, the agricultural area of organic farming in Spain is around 2.44 million hectares, just over 45% of which is located in Andalusia [33]. Spain is the world's leading producer of organic almonds. The area of almond crop certified in organic management in Spain is 166 thousand hectares, of which about 72 thousand hectares are in Andalusia [33], which represents 37% of the total area of almonds in production in this region. Most of the organic almonds are in marginal areas with low intensive production systems, and only 21% of the cultivated area has support irrigation [34]. One of the factors that has increased interest in ecological management is that it is seen by farmers as a viable option to increase the profitability of their orchards. This is due to the growing demand, the price differential compared to conventional almonds, and to the fact that the main producing countries worldwide (USA and Australia) find it complicated to implement organic farming [35].

However, organic agriculture is one of the strategies that can be used to mitigate the negative environmental effects of conventional agriculture. The organic system must be accompanied by other soil conservation strategies, such as reduction or suppression of the tillage or the use of cover crops, aiming to encourage the soil health and quality [36,37]. The most common soil management system in almond orchards is conventional tillage. However, this practice is becoming less frequently used, since it implies a series of negative

effects with great impacts on soil quality [38]. In the long-term conventional tillage causes the compaction of the soil [39], the degradation of soil structure [40], high erosion rates, and carbon and nutrient losses [41–43]. In this context, the urgency to implement sustainable soil conservation measures is widely accepted, and it is vital to ensure sustainability of agroecosystems, especially under conditions of climate change [19,37,44]. In rainfed almond orchards, the implementation of soil conservation measures promotes various benefits such as the reduction of water erosion of soil, improvement in the provision of ecosystem services, the increase in soil organic carbon (SOC) sequestration, and the stimulus of wild pollinators, among others [45–48].

On the other hand, according to Gervasi et al. [49], almond is the most consumed tree nut worldwide because of its potential benefits to human health. Almonds and other tree nuts have healthy nutritional profiles. Studies on the almond composition have shown that it has a high content of bioactive compounds such as fatty acids, lipids, amino acids, proteins, carbohydrates, dietary fibers, phytosterols, polyphenols, vitamins, and minerals, as well as other secondary metabolites [50,51]. In this sense, there is a large number of studies suggesting that the consumption of almonds reduces the risk of cardiovascular disease, improves diet and contributes to satiety, improves the inflammatory processes associated with metabolic syndrome and type 2 diabetes, has prebiotics and antioxidant functions, can contribute to the prevention of cancer and cellular stress, and can act as immuno-modulator [52–55]. The nutrient composition of almonds depends mainly on the genotype, although it can also be influenced by other factors, such as edaphoclimatic conditions, the growing region, cultivation practices, grain maturity, etc., or the interactions of these factors [51,56].

Although the market for organic products is still small, the demand for organic food products is increasing worldwide. According to Sahota [57], the international sales of organic food and drink reached USD 112 billion in 2019, and the market has expanded by 55% since 2013. The main reasons for consumers to buy organic food products are because they are perceived as healthier, safer, more nutritious, and respectful of the environment and animal welfare [58–60].

The main objective of this study in an organic rainfed production system was to evaluate the impact of soil conservation practices (no tillage or legume cover) on select physical, chemical, and biological soil properties, as well as their implications for yield and almond quality in the Mediterranean region.

## 2. Materials and Methods

### 2.1. Study Area

The study was conducted during 2016–2021 on an organic almond orchard located in Lanjarón (Granada, SE Spain) with UTM coordinates of X = 456,720.423; Y = 4,083,607.192. The soils in the experimental area are Typic Xerorthens [61] with a sandy loam texture. The climate is temperate with a hot, dry summer and a cold-temperate winter according to the Koppen climate classification. The average annual temperature is 15 °C, and the average annual rainfall is 480 mm.

Table 1 displays the textural composition of soils under different soil management strategies (SMSs) used in the present experiment.

**Table 1.** Soil texture and composition at different depths for experimental plots under soil management strategies.

| SMS | Soil Depth | | | | | | | |
|-----|------------|------|------|-------|------|------|------|-------|
| | 0–10 cm | | | | 10–25 cm | | | |
| | Sand | Clay | Silt | | Sand | Clay | Silt | |
| | (%) | | | Class | (%) | | | Class |
| TT | 47 | 21 | 32 | Loam | 52 | 22 | 26 | Sandy clay loam |
| NT | 57 | 16 | 27 | Sandy loam | 56 | 18 | 26 | Sandy loam |
| VF | 58 | 17 | 25 | Sandy loam | 59 | 17 | 24 | Sandy loam |
| VS | 49 | 21 | 30 | Loam | 50 | 26 | 24 | Sandy clay loam |
| VS-VE | 59 | 21 | 20 | Sandy clay loam | 64 | 22 | 14 | Sandy clay loam |

SMS, soil management strategy; TT, traditional tillage; NT, no tillage; VF, cover of *Vicia faba*; VS, cover of *Vicia sativa*; VS-VE, cover of *Vicia sativa* and *Vicia ervilia*.

## 2.2. Experimental Design

The experimental plots used were part of a 65-year-old rainfed plantation of almond trees (*Prunus dulcis* (Mill.) D.A. Webb) spaced $7 \times 7$ m (~200 trees ha$^{-1}$), of one of the most commonly cultivated varieties in Spain, namely, "Marcona".

The experimental plots were duplicated twice for each of the soil management strategies studied and randomly distributed in the study area (Figure 1).

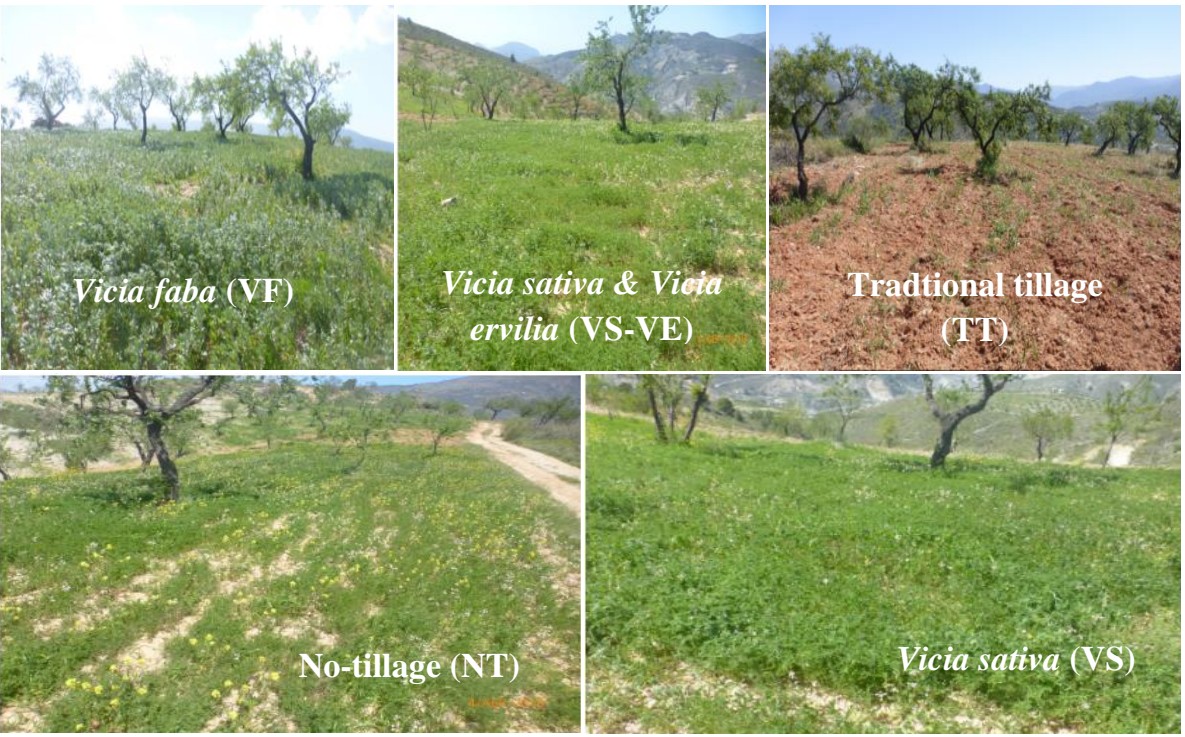

**Figure 1.** Soil management strategies implemented in the study.

Table 2 shows the soil management strategies applied to each agricultural production system during the 5-year monitoring period.

**Table 2.** Soil management strategies applied in almond organic production system during study period (2016–2021).

| Date | Soil Management Strategies (SMS) | | | | |
|---|---|---|---|---|---|
| | TT | NT | VF | VS | VS-VE |
| January–February | Moldboard tillage (30 cm) | —— | —— | —— | —— |
| March [a] | Fungicidal treatment (Cu oxychloride; 150 L/ha) | Fungicidal treatment (Cu oxychloride; 150 L/ha) | Fungicidal treatment (Cu oxychloride; 150 L/ha) | Fungicidal treatment (Cu oxychloride; 150 L/ha) | Fungicidal treatment (Cu oxychloride; 150 L/ha) |
| April | Tillage | Mechanical vegetation control | Disc harrow incorporation | Disc harrow incorporation | Disc harrow incorporation |
| April [a] | Insecticide treatment (K soap + pyrethrins; 200 L/ha) | Insecticide treatment (K soap + pyrethrins; 200 L/ha) | Insecticide treatment (K soap + pyrethrins; 200 L/ha) | Insecticide treatment (K soap + pyrethrins; 200 L/ha) | Insecticide treatment (K soap + pyrethrins; 200 L/ha) |
| August–September | Almond harvest | Almond harvest | Almond harvest | Almond harvest | Almond harvest |
| September–October [b] | —— | —— | Sowing *V. faba* (150 kg/ha) | Sowing *V. sativa* (100 kg/ha) | Sowing *V. sativa* + *V. ervilia* (100 kg/ha) |
| October | Fertilization [c] (NK (Ca) 3-9(8); 2 kg/tree) | Fertilization (NK (Ca) 3-9(8); 2 kg/tree) | Fertilization (NK (Ca) 3-9(8); 2 kg/tree) | Fertilization (NK (Ca) 3-9(8); 2 kg/tree) | Fertilization (NK (Ca) 3-9(8); 2 kg/tree) |
| October | Thinning pruning | Thinning pruning | Thinning pruning | Thinning pruning | Thinning pruning |

TT, traditional tillage; NT, no tillage; VF, cover of *Vicia faba*; VS, cover of *Vicia sativa*; VS-VE, cover of *Vicia sativa* and *Vicia ervilia*. [a] Only in the last two years; [b] before the first autumn rains; [c] with an organic fertilizer from animal manure and certified by the Andalusian Committee for Organic Agriculture (CAAE).

### 2.3. Soil Sampling and Analysis

For the determination of physico-chemical properties, two undisturbed soil cores (~100 cm$^3$) (soil sampling ring kit—model C53, Eijkelkamp) and two disturbed samples were taken per experimental plot at two depths (0–10 and 10–25 cm depth) at the beginning (autumn 2016) and at the end of the study (June 2021) (5 SMSs × 2 plots × 2 replicates × 2 depth × 2 dates = 80 disturbed samples and 80 undisturbed samples in total). The sampling was always carried out in the central part of each experimental plot to avoid the edge effect and randomly distributed on the surface of the plot. The disturbed soil samples were sieved (2 mm) and air-dried for later chemical analysis. For the determination of soil microbial activity, three disturbed soil samples were taken at two depths per treatment (0–10 cm and 10–25 cm of depth) (5 SMSs × 3 replicates × 2 depth = 30 samples in total). The samples were sieved (2 mm) and stored at 4 °C in plastic bags loosely tied to assure sufficient aeration and to prevent moisture loss until microbial activity was analyzed.

#### 2.3.1. Soil Physical Properties

Particle size distribution was determined by the Boyoucos densimeter [62]. For bulk density, undisturbed soil samples were taken in rings of known volume and 5 cm in diameter and height [63]. Bulk density was obtained by dividing oven-dried (105 °C) soil weight by the soil sample volume.

Water content values were determined on crushed samples passed through a 2 mm sieve and placed in a 5.5 cm ring on 33 and 1500 kPa pressure plates. Gravimetric water content was determined using a pressure plate apparatus [64]. At each pressure, soil samples were equilibrated for 48 h, weighed, and returned to the plate extractor for the next pressure step. At the end, soil samples were oven-dried at 105 °C for 24 h to determine by weighing the water soil content. The points calculated were 33 (field capacity) and 1500 kPa (permanent wilting point).

The stability of soil aggregates was characterized by the distribution of the size of the aggregates in water through an automated sieving technique (FRITSCH Vibratory Sieve Shaker Analysette 3 Pro). This system can control the parameter of the agitator and the flow of water through the sieves, eliminating the influence of the operator [65]. The stability of the water aggregates was measured by placing 100 g of soil on top of a stack of four sieves

with different mesh sizes, namely, 1, 0.5, 0.25, and 0.05 mm, resulting in four kinds of sizes of aggregate: large aggregates (>1 mm), macro-aggregates (0.25–1 mm), micro-aggregates (50–250 μm), and the lime fraction + clay (<50 μm) considered not added. For this study, the first two fractions were grouped together as the total of macro-aggregates.

The mean weighted diameter (*MWD*) was calculated by multiplying the proportion of soil in each of the aggregate size classes by the mid-point of each size class.

$$MWD = \sum_{i=1}^{n} Ai\ Pi \tag{1}$$

where *A*, mean diameter of each size fraction *i* (mm); *P*, the proportion of aggregates obtained for each size class; *n*, number of fractions.

Sand correction was performed to account for soil texture differences, as described by Six et al. [66], to avoid overestimation due to sand weight. The aggregate stability index (ASI) was calculated as the ratio between the *MWD* of water stable aggregates to the *MWD* of dry aggregates [67].

### 2.3.2. Soil Chemical Properties and Microbial Activity

Soil pH and electrical conductivity were determined in the 1:2.5 soil/water extract. The method described by Hense [68] was used to determine Kjeldahl-N. Available P was determined using the Olsen et al. [69] method, available K according to the standard methods [70], and the estimation of soil organic carbon (SOC) by means of organic matter determined by the Walkley and Black [71] method and using the van Bemmelen factor for the calculations. The extractable elements (Cu, Zn, Fe, and Mn) were determined using the DTPA (diethylenetriaminepentaacetic acid) method [72]. The concentration of the elements was performed using the technique of inductive coupling plasma (ICP).

For the measurement of total soil microbial activity, a colorimetric enzymatic method, fluorescein diacetate hydrolase (FDA), was used [73]. Fluorescein diacetate (3′6′-diacetyl-fluorescein) is a fluorescein conjugated with two acetate radicals. This colorless compound is hydrolyzed by both exoenzymes and membrane-bound enzymes, releasing a final colored product, fluorescein. The results were expressed as μg fluorescein $g^{-1}$ of soil, since this chemical compound is generated by hydrolysis of lipases and esterases from living microorganisms [74]. This is a simple, fast, and sensitive method that has extensive acceptance as a measure of total microbial activity in a wide range of soil, and that has been identified as an early warning indicator of soil quality change [75]. The activity of FDA represents the hydrolytic activity of several fungal and bacterial enzymes and can be interpreted as an index of the potential for release of organic nutrients from organic matrices [76].

### 2.4. Foliar Sampling and Analysis

Sampling was carried out in June during the last two seasons of the monitoring period on fully developed new leaves from the middle parts of shoots of the year and taken in the 4 orientations (N, S, W, E). Four samples were taken for each of the soil management strategies (5 SMSs × 4 repetitions = 20 samples). All plant samples were prepared via the microwave digestion system (Novawave, SCP Science Inc, Baie-d'Urfé, QC, Canada) using a mixture of $HNO_3$ and HCl [77]. All analyses were performed on an Avio 200 ICP-OES system (PerkinElmer Inc, Waltham, MA, USA).

### 2.5. Yield and Physico-Chemical Analysis of Almonds

At the end of each season, the almond yield under different SMSs was evaluated by weighing the almond kernel production after removal of the pericarp. The production of each tree was measured, calculating the average production per tree for each SMS and using this value to calculate the production per area. For the physical–chemical analysis, a composite sample was taken from the central trees of each experimental plot. For this, about 3 kg of in-shell almonds (per SMS) were sent to Miguel Hernandez University (Orihuela, Alicante, Spain) for quality analysis.

2.5.1. Moisture Content, Water Activity, Textural Analysis, and Instrumental Color

The moisture content of almonds according to the AOAC standard method was determined. For this purpose, 2 g of grinded sample was placed in metallic trays and dried in an oven (60 °C) to constant mass [78]. A water activity meter (Novasina AW-SPRINT TH500; Pfaffikon, Zurich, Switzerland) for determination of the water activity ($a_w$) was used. The measurement for each sample was conducted in triplicate, and the reported data represent the average.

The texture of almonds was determined by the using a texture analyzer (Stable Micro Systems, model TA-XT2i, Godalming, UK) loaded with a 30 kg cell and a probe (Volodkevich Bite Jaw HDP/VB). The trigger was set at 15 g, and the test displacement rate was 1 mm s$^{-1}$ over a specified distance of 3 mm. The texture of almonds was analyzed by five parameters: fracturability (mm), hardness (N), work done to shear (Ns), average force (N), and number of fractures (peaks count). Data are reported as the mean value of twenty measurements.

A Minolta CR-300 colorimeter (Minolta, Osaka, Japan) was used to evaluate the color of almond kernels. As references, a D65 illuminant and a 10° observer was used. The color was provided as CIEL*a*b* coordinates defining the color in a three-dimensional space, and it was expressed in three numerical values. Twenty measurements were taken on each sample, and values for L*, a*, and b* were averaged.

2.5.2. Organic Acids and Sugars

Organic acid and sugar profiles using high-performance liquid chromatography (HPLC) equipment were determined. The extraction consisted of the homogenization of 1 g of ground almonds (10 s in a Moulinex grinder, AR110830) with 5 mL of phosphate buffer (pH = 7.8). The obtained samples were filtered (0.45 μm Millipore membrane filter) and injected into a Hewlett Packard (Wilmington, DE, USA) series 1100 HPLC. As elution buffer, 0.1% ortophosphoric acid was used. For separation of compounds, a Supelcogel TM C-610H column (30 cm × 7.8 mm) with a pre-column (Supelguard 5 cm × 4.6 mm; Supelco, Bellefonte, PA, USA) was used. Sugars were detected by a refractive index detector (RID) and organic acids by a diode-array detector (DAD) through absorbance measurement at 210 nm. Analyses were triplicated, and results were expressed as g kg$^{-1}$ dry weight.

2.5.3. Determination of Total Polyphenols and Antioxidant Activity

Ground almonds were used for the extraction of polyphenols and antioxidant activity using a methanolic extractant (MeOH:$H_2O$; 80:20; *v*:*v* + 1%HCl). Total polyphenol content (TPC) was determined using the Folin–Ciocalteu colorimetric method [79]. Almond extract (0.1 mL) was mixed with Folin–Ciocalteu reagent (0.2 mL) and 2 mL of distilled water. After 3 min, 1 mL of a 20% aqueous solution of sodium carbonate ($Na_2CO_3$) was added. The absorbance at 765 nm was measured after 1 h. The results were expressed as g of gallic acid equivalents (GAE) kg$^{-1}$. Data are reported as the mean value of three repetitions.

The antioxidant activity of the almonds was measured using ABTS$^{\bullet+}$ (2,2′-azino-bis(3-ethylbenzthiazoline-6-sulphonic acid) according to Re et al. [80], and DPPH$^{\bullet}$ (2,2-diphenyl-1-picryl-hydrazyl-hydrate) radicals described by Brand-Williams et al. [81] were determined. Moreover, antioxidant activity was also determined by the ability to reduce iron ions (FRAP) according to Benzie and Strain [82]. The measurements of antioxidant activities were carried out according to detailed protocol described by Lipan et al. [83]. Antioxidant capacity was expressed as mmol of Trolox kg$^{-1}$. Data were reported as the mean value of four determinations. All measurements were carried out in an ultraviolet–visible (UV-vis) spectrophotometer (Helios Gamma model, UVG 1002E; Helios, Cambridge, UK).

2.5.4. Fatty Acids Analysis

The fatty acids methyl esters (FAMEs) were determined using methylation as previously described by Lipan et al. [84] with some modification. Methyl esters of fatty acids

were separated in a Shimadzu GC17, a gas chromatography coupled with a flame ionization detector and a DB-23 capillary column. As the carrier gas, He was used with an initial flow rate of 1.2 mL min$^{-1}$, while the detector gasses were H$_2$ (30 mL min$^{-1}$), air (350 mL min$^{-1}$), and He (35 mL$^{-1}$) as make-up gas. The injector and detector temperatures were 250 and 260 °C, respectively, with an injection volume of 0.6 uL and a split ratio of 1:10. The preliminary temperature was 175 °C for 10 min, a temperature gradient of 3 °C min$^{-1}$ until 215 °C, and maintaining 215 °C for 15 min. The identification of FAME peaks was conducted via comparison with the retention times of the standards (FAME Supelco MIX-37). Results were expressed quantitatively as g kg$^{-1}$ using methyl nonadecanoate as an internal standard.

*2.6. Statistical Analysis*

Data were checked for normality and homoscedasticity, and analysis of variance (ANOVA) was applied using STATGRAPHICS Centurion XV Windows version 15.1 (Stat-Point, Inc., Herndon, VA, USA). Treatment means were separated by the Tukey Multiple Range Test at 5% level of significance. A Pearson correlation matrix was constructed among the soil properties studied.

## 3. Results and Discussion

*3.1. Soil Physical Properties*

Figure 2 shows the soil bulk density ($\rho b$) and available water capacity (AWC) during the study period from the beginning to the end of the experiment. At the end of the study there were no significant differences among SMSs for bulk density ($\rho b$) in the upper layer (0–10 cm). A significant change for $\rho b$ between the beginning and the end of the study in the upper soil layer (0–10 cm) was registered in VS-VE with a decrease of 8.1%. NT and VF treatments also registered not significant decreases in $\rho b$ of 3.1 and 0.4%, respectively. In this soil layer, the accumulation of crop residues and SOC can lead to a decrease in $\rho b$, as was stated by Kay and VandenBygaart [85]. Sometimes the amount of residue is not enough to improve SOC levels, which tends to improve soil structure and increases porosity. In these cases, $\rho b$ may be higher in conservation agriculture than in conventional tillage [86]. In addition, it must be taken into consideration that the effects of the conservation techniques on soil physical properties such as $\rho b$ are not immediate, and it is necessary that a time elapses from the conversion to this system [87]. In addition, soil with a higher percentage of sand usually has higher $\rho b$ and lower AWC than those with fine silts and clays, and this may account for the initial differences because TT and VS showed lower sand contents than all the other management strategies (Table 1).

Regarding the available water capacity (AWC) in the upper soil layer, it increased in all SMSs, especially in VS-VE and NT with more than 21%, although there are no important differences among them. From soil depths of 10–25 cm, all treatments increased AWC, highlighting VS-VE with 23%.

There are numerous studies that have reported greater water availability in conservation agriculture systems compared to conventional tillage [88–90] due to the fact that residue retention and cover crops improve infiltration and reduce losses from evaporation and runoff. This improvement in the availability of soil moisture is especially important to maintain crop yields in rainfed systems in the face of the challenge of water scarcity in a climate change scenario [91]. In this sense, in rainfed conditions, the proper management of cover crops is crucial to prevent competition for resources (water and nutrients) with the main crop that can result in a decline in productivity [92–94].

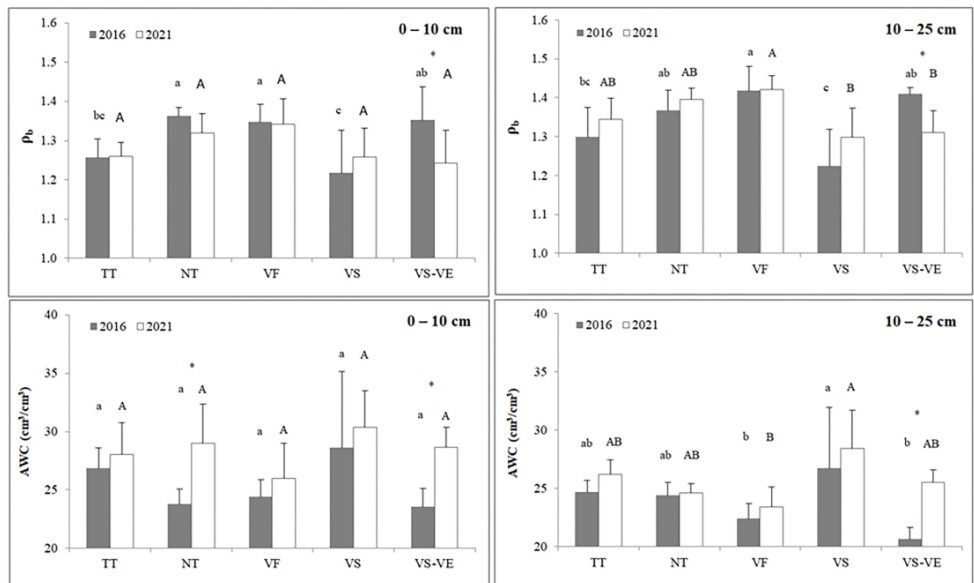

**Figure 2.** Average soil bulk density (ρb) and available water capacity (AWC) by effect of different soil management strategies. TT, traditional tillage; NT, no tillage; VF, cover of *V. faba*; VS, cover of *V. sativa*; VS-VE, cover of *V. sativa* and *V. ervilia*. Numerical values are means, bars are standard deviation. Different lowercase letters indicate significant differences between management strategies at the beginning of the study; different capital letters indicate significant differences between management strategies at the end of the study; *, indicates significant differences in management strategy between the two periods by Tukey's test, $p < 0.05$.

Figure 3 shows the dry and wet mean weight diameter and aggregate stability index for the upper soil layer (0–10 cm) by the effect of different soil strategies applied at the end of the study period.

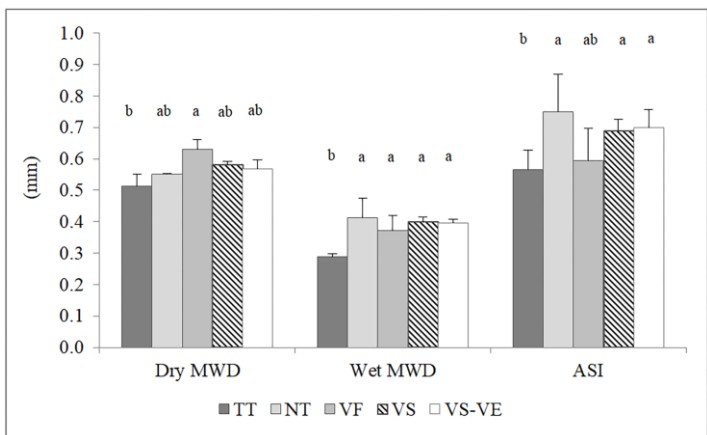

**Figure 3.** Dry and wet mean weight diameter (*MWD*) and aggregate stability index (ASI) by effect of different soil management strategies. TT, traditional tillage; NT, no tillage; VF, cover of *V. faba*; VS, cover of *V. sativa*; VS-VE, cover of *V. sativa* and *V. ervilia*. Bars are standard deviation. Different letters indicate significant differences between management strategies (Tukey's test, $p < 0.05$).

The aggregate stability at soil depths 0–10 cm was greater for conservation agriculture practices (NT, VF, VS, and VS-VE) than for traditional tillage (TT). This coincides with the results of other studies that have shown the effectiveness of reducing and/or eliminating tillage and the use of plant cover in improving aggregate stability [95,96]. That is, tillage promotes the breakdown of soil macro-aggregates, which affects the stability of soil aggregates while increasing the mineralization of SOC retained in the macro-aggregates. Therefore, the

presence of large amounts of crop residues at the soil surface and minimal soil disturbance increase the proportion of large aggregates in the soil, improving the soil microbial stability, the organic matter stabilization into the aggregates, and the resistance of the soil structure to water breakage, as was reported by Al-Kaisi et al. [97] and Belmonte et al. [95].

*3.2. Soil Bio-Chemical Properties*

Statistically significant differences for the pH values at 0–10 cm between VS and VS-VE with the rest of the SMSs were found at the beginning of the study (Figure 4). In the same way, at final sampling, the highest pH was registered for VS and VS-VE, although for both soil treatments there was a decrease in pH at the final sampling of 4.3 and 3.2%, respectively, this difference being statistically significant only for VS. There was also a slight decrease for TT ($-0.43\%$), while the rest of treatments increased the pH values, although the difference was statistically significant only for VF and VS. The pH at 10–25 cm depths, at the beginning of the study, also presented statistically significant differences among SMSs (NT, VF) < (TT, VS-VE) < (VS). At the end of the experiment in 2021, the highest pH values continued to be recorded for VS and VS-VE, although both suffered a decreasing trend with respect to the initial sampling in 2016 (5.3% and 1.2, respectively). The pH for the TT showed a slight decrease, while an increasing trend was found for the rest of the strategies. The changes between both periods were statistically significant only for VF and VS. The difference in soil pH dynamics among treatments may be due to the differences in the initial pH, a factor that has a substantial impact on the change in soil pH through the incorporation of residues, since it affects the mineralization and the decomposition rate of organic compounds [98].

In addition, the decrease of soil pH in the VS and VS-VE may be due to a greater accumulation of soil organic matter in the upper soil layers and release of organic acids upon decomposition in the soil surface [99,100].

The electrical conductivity (EC) of the soil-saturation extract values at both sampled soil depths were much lower than the critical level of 4.0 dS m$^{-1}$ for all SMSs, indicating a minimal threat of salt toxicity for plants (Figure 4). According to Husson et al. [101], conservation agricultural practices decreased soil EC when it was initially high and increased it when it was low. In this line, in a sorghum–wheat system, the EC of soil-saturation extract was significantly affected by different tillage, saline irrigation, and mulch practices. Instead, according to Roldán et al. [102], the soil EC in a maize crop was not affected by soil management when comparing no-till with the addition of plant residues, no-till with cover crops, and conventional tillage. Concretely, Soni et al. [103] reported that among tillage practices, zero tillage recorded lower EC (5.5 dS m$^{-1}$) compared to reduced tillage and conventional tillage (6.66 and 7.09 dS m$^{-1}$, respectively). Those practices that increase the retention of plant residues on the soil surface promote an improvement in infiltration rates and decrease the soluble salt contents in the soil surface, reducing soil EC values [104]. In contrast, in our study, EC recorded a 30% increase in VS-VE, which coincides with the results of Demir et al. [105] who registered an increase in EC in a study with cover crops in apricot.

In relation to SOC content in the upper soil layer, all SMSs applied increased their values of SOC (Figure 4). At the end of the study in 2021, statistically significant differences among VF with VS and VS-VE treatments were found. Similarly, for soil depths 10–25 cm, all SMSs also registered an increase in SOC stocks. It is well known that soil tillage increases the decomposition rates of organic matter by the alteration of soil structure and the organic matter retained in soil aggregates [106]. In this sense, conservation tillage systems (mainly no-tillage and minimum tillage) increase the accumulation of SOC in the soil surface through the reduction of their losses (by oxidation and/or erosion), the increase in organic carbon inputs to the soil (plant residues), or a combination of both factors [107,108]. Many studies have reported that the SOC was higher in soils with plant cover in woody fruit orchards compared to conventional tillage [109–111]. The lowest SOC content for both depths was recorded in plots with VF cover, although an increasing trend during the study period was evident. This could be due to the fact that this was the plot with the lowest

initial SOC content, and there is evident that the cultivation implementation of *V. faba* cover encouraged improvements in SOC, although the intensity was not comparable to that of the remaining SMSs.

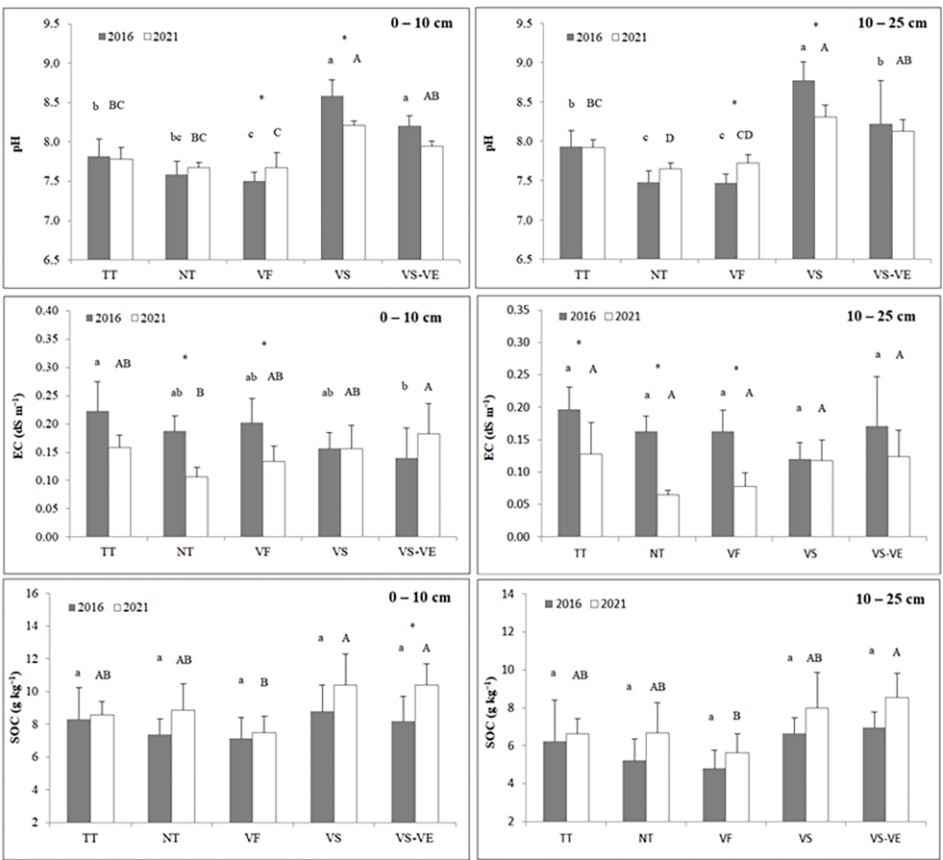

**Figure 4.** Average pH, electrical conductivity (EC) of the soil-saturation extract, and soil organic carbon (SOC) by the effect of different soil management strategies. TT, traditional tillage; NT, no tillage; VF, cover of *V. faba*; VS, cover of *V. sativa*; VS-VE, cover of *V. sativa* and *V. ervilia*. Bars are standard deviation. Different lowercase letters indicate significant differences at the beginning of study; different capital letters indicate significant differences at the end of the study; *, indicates significant differences between the two sampling periods (Tukey's test, $p < 0.05$).

3.2.1. Soil Macro and Micronutrients

Figure 5 shows the NPK soil contents subjected to different soil management strategies during the five-year monitoring period. Particularly, for the superficial soil layer (0–10 cm), there was an increase in N content for SMSs with the cover crop and for VS and VS-VE at soil depths of 10–25 cm, decreasing in the rest of the strategies. In this line, Ranaivoson et al. [112] concluded that the decomposition of legume crop cover generally promotes an increase in soil mineral N. In this sense, Durán et al. [109], in hillslope farming with rainfed olive orchards in southwest Spain, showed an increase in soil nitrogen with minimum tillage and legume strips compared with conventional tillage. In a long-term experiment, Mazzoncini et al. [113] showed that in the soil surface layer, legume cover crops significantly increased soil total nitrogen concentration compared with the system without cover crops. That is, cover crops based on leguminous plants in the organic systems have the potential to fix and supply the plant nutrients required for their own growth as well as transfer those nutrients to almond trees.

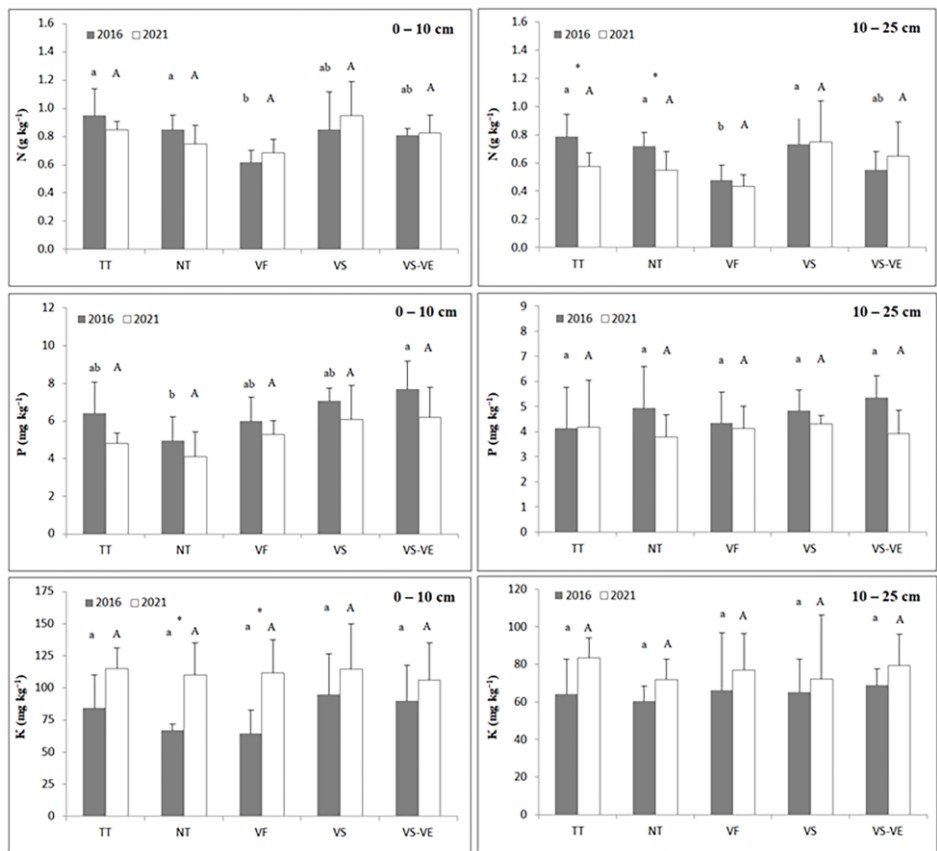

**Figure 5.** Average nitrogen (N), phosphorus (P), and potassium (K) contents by effect of different soil management strategies. TT, traditional tillage; NT, no tillage; VF, cover of *V. faba*; VS, cover of *V. sativa*; VS-VE, cover of *V. sativa* and *V. ervilia*. Bars are standard deviation. Different lowercase letters indicate significant differences at the beginning of the study; different capital letters indicate significant differences at the end of the study; *, indicates significant differences between the two sampling periods (Tukey's test, $p < 0.05$).

In relation to P content, all SMSs showed a decreasing trend without statistically significant differences among treatments (Figure 5). In this context, many studies have shown a lower availability of P in soils with plant cover, which is probably due to the immobilization of this nutrient in the biomass of the cover crop and/or a greater retention in the solid phase of the soil [114,115]. On the contrary, all the SMSs registered an increase in assimilable K, without statistically significant differences between treatments at the end of the study in 2021, due to the high variability of the data. Similar findings regarding available soil K have been reported by Durán et al. [109] in organic olive orchards with legume cover crops compared with conventional system. In this line, Sujatha et al. [116] highlighted that the extensive root systems of legumes foster them to release organic acids from their roots, which augments available K in soils.

In general, the effect of soil conservation practices on nutrient availability depends on various factors such as the nature of crop residues, the amount of surface residues, or the proportion of soil covered by them [112,117]. Soil conservation techniques involve more provision of plant residues, increasing the input of organic material into the soil that undoubtedly can lead to increased plant nutrient stores. However, this does not always translate into greater plant nutrient availability [118]. This can occur due to slower rates of mineralization, greater rates of immobilization, and/or the consumption and retention by cover crops [119,120]. In this case, it may be necessary to optimize the organic fertilizer doses to maintain the yield.

Table 3 shows the main soil micronutrient contents at the beginning and end of the experiment. The contents for all micronutrients were higher at the upper soil layer than at 10–25 cm. This was probably due to the increase of plant residue and its accumulation in the surface layer and the lack of soil mixing in conservation practices. At the end of the study, the lowest concentration of available Fe in the surface layer was recorded in TT with statistically significant differences with respect to treatments with cover crops, which had the highest concentrations. Similarly, TT was also the soil management strategy with the lowest final Cu concentration at soil depths 0–10 cm, with statistically significant differences only for VS-VE plots. In relation to Zn, the highest concentrations in the surface layer were recorded in the SMSs with cover crops of VF and VS, although there are no statistically significant differences among them (Table 3). These results are in line with numerous studies that have shown how conservation agriculture practices improve the availability of micronutrients compared to conventional practices [121–123].

**Table 3.** Average soil micronutrient contents by effect of different soil management strategies during the study period.

| Micronutrients/Sampling Year | Soil Management Systems | | | | |
| --- | --- | --- | --- | --- | --- |
| | **TT** | **NT** | **VF** | **VS** | **VS-VE** |
| | **(mg kg$^{-1}$)** | | | | |
| Fe | | | 0–10 cm | | |
| 2016 | 1.66 ± 0.59 b | 1.84 ± 0.68 b | 2.04 ± 0.51 b | 2.70 ± 0.80 ab | 3.47 ± 0.62 a |
| 2021 | 1.53 ± 0.29 c | 2.20 ± 0.65 bc | 4.84 ± 1.06 a | 4.03 ± 1.32 ab | 4.87 ± 0.42 a |
| | n.s. | n.s. | * | n.s. | * |
| Cu | | | | | |
| 2016 | 0.21 ± 0.07 a | 0.11 ± 0.06 a | 0.21 ± 0.11 a | 0.13 ± 0.05 a | 0.19 ± 0.17 a |
| 2021 | 0.30 ± 0.04 b | 0.35 ± 0.14 b | 0.46 ± 0.10 ab | 0.59 ± 0.24 ab | 0.68 ± 0.17 a |
| | n.s. | * | * | * | * |
| Mn | | | | | |
| 2016 | 3.77 ± 0.47 ab | 4.70 ± 0.81 a | 3.23 ± 0.74 b | 4.32 ± 1.25 ab | 4.83 ± 0.86 a |
| 2021 | 4.99 ± 0.72 a | 3.62 ± 1.07 ab | 3.26 ± 0.80 b | 3.54 ± 0.73 ab | 3.15 ± 0.68 b |
| | * | n.s. | n.s. | n.s. | * |
| Zn | | | | | |
| 2016 | 3.07 ± 0.43 a | 2.93 ± 0.94 a | 3.50 ± 0.89 a | 3.32 ± 0.60 a | 2.92 ± 1.06 a |
| 2021 | 2.61 ± 0.16 a | 2.74 ± 0.24 a | 2.96 ± 0.22 a | 3.58 ± 1.00 a | 3.78 ± 0.94 a |
| | n.s. | n.s. | n.s. | n.s. | n.s. |
| | | | 10–25 cm | | |
| Fe | | | | | |
| 2016 | 1.37 ± 0.45 b | 1.69 ± 0.32 b | 1.80 ± 0.50 b | 2.19 ± 0.59 ab | 2.81 ± 0.92 a |
| 2021 | 1.83 ± 0.21 b | 1.74 ± 0.26 b | 3.93 ± 1.18 a | 2.86 ± 1.21 ab | 3.32 ± 1.03 ab |
| | n.s. | n.s. | * | n.s. | n.s. |
| Cu | | | | | |
| 2016 | 0.12 ± 0.03 a | 0.08 ± 0.04 a | 0.12 ± 0.05 a | 0.09 ± 0.01 a | 0.14 ± 0.05 a |
| 2021 | 0.23 ± 0.07 b | 0.51 ± 0.07 a | 0.37 ± 0.10 ab | 0.36 ± 0.15 ab | 0.50 ± 0.17 a |
| | * | * | * | * | * |
| Mn | | | | | |
| 2016 | 2.43 ± 0.38 b | 3.81 ± 0.48 a | 1.97 ± 0.72 b | 2.80 ± 0.97 ab | 3.08 ± 0.59 ab |
| 2021 | 3.40 ± 0.91 a | 2.06 ± 0.50 b | 2.15 ± 0.49 b | 2.63 ± 0.75 ab | 2.02 ± 0.40 b |
| | * | * | n.s | n.s. | * |
| Zn | | | | | |
| 2016 | 1.63 ± 0.56 a | 1.89 ± 0.72 a | 2.40 ± 0.87 a | 1.54 ± 0.25 a | 1.40 ± 0.42 a |
| 2021 | 2.09 ± 0.09 b | 2.27 ± 0.23 ab | 2.39 ± 0.37 ab | 2.31 ± 0.30 ab | 3.13 ± 0.88 a |
| | n.s. | n.s. | n.s. | * | * |

TT, traditional tillage; NT, no tillage; VF, cover of *V. faba*; VS, cover of *V. sativa*; VS-VE, cover of *V. sativa* and *V. ervilia*. Means ± standard deviation. Different letters in the same line indicate significant differences between management (Tukey's test, $p < 0.05$). *, significant differences between both periods; n.s., no significant differences between periods.

According to Beltrán et al. [124] and Khoshgoftarmanesh et al. [125], plant micronutrient accumulation in the topsoil under conservation agriculture techniques is due to their incorporation through plant residues on the soil surface. The Mn content showed an inverse trend compared to those described for Fe, Zn, and Cu, the lowest concentrations being detected for SMSs with cover crops of VF, with statistically significant differences in VF and VS-VE with TT. This decrease in available Mn in cover crop treatments may be due to the removal of large amounts of Mn during the growth of cover plants, as was highlighted by Gao et al. [126] or the Mn content from plant cover residues that greatly influenced its decomposition rates [127].

Table 4 displays a pooled correlation matrix among the soil physico-chemical properties at the end of the study for each soil management strategy studied.

**Table 4.** Correlation matrix of different soil parameters, irrespective of depth, for each SMS.

| | ρb | AWC | pH | EC | SOC | Na | K | Mg | P | N | Fe | Cu | Mn | Zn |
|---|---|---|---|---|---|---|---|---|---|---|---|---|---|---|
| | | | | | | | Traditional tillage | | | | | | | |
| ρb | 1 | | | | | | | | | | | | | |
| AWC | −0.790 ** | 1 | | | | | | | | | | | | |
| pH | 0.632 * | −0.755 ** | 1 | | | | | | | | | | | |
| EC | −0.746 ** | 0.675 * | −0.805 ** | 1 | | | | | | | | | | |
| SOC | −0.809 ** | 0.405 | −0.412 | 0.649 ** | 1 | | | | | | | | | |
| Na | −0.201 | 0.136 | 0.275 | −0.100 | 0.187 | 1 | | | | | | | | |
| K | −0.505 | 0.145 | −0.168 | 0.211 | 0.840 *** | 0.202 | 1 | | | | | | | |
| Mg | −0.453 | 0.135 | 0.237 | −0.042 | 0.530 | 0.374 | 0.566 | 1 | | | | | | |
| P | −0.389 | 0.299 | −0.179 | 0.055 | 0.285 | 0.259 | 0.318 | 0.689 * | 1 | | | | | |
| N | −0.520 | 0.378 | −0.647 * | 0.485 | 0.688 * | −0.062 | 0.750 ** | −0.038 | 0.025 | 1 | | | | |
| Fe | 0.496 | −0.548 | 0.925 *** | −0.753 ** | −0.396 | 0.461 | −0.190 | 0.419 | 0.134 | −0.732 ** | 1 | | | |
| Cu | −0.114 | 0.052 | −0.241 | 0.217 | 0.433 | 0.204 | 0.582 | −0.256 | −0.368 | 0.787 ** | −0.405 | 1 | | |
| Mn | −0.862 *** | 0.711 ** | −0.575 | 0.541 | 0.765 ** | 0.326 | 0.659 * | 0.602 | 0.722 ** | 0.563 | −0.345 | 0.149 | 1 | |
| Zn | −0.613 | 0.190 | −0.473 | 0.414 | 0.798 ** | 0.014 | 0.802 ** | 0.283 | 0.314 | 0.828 ** | −0.517 | 0.484 | 0.662 * | 1 |
| | | | | | | | No Tillage | | | | | | | |
| ρb | 1 | | | | | | | | | | | | | |
| AWC | −0.848 *** | 1 | | | | | | | | | | | | |
| pH | −0.281 | 0.039 | 1 | | | | | | | | | | | |
| EC | −0.482 | 0.377 | 0.085 | 1 | | | | | | | | | | |
| SOC | −0.631 * | 0.516 | −0.273 | 0.671 * | 1 | | | | | | | | | |
| Na | −0.301 | 0.455 | −0.110 | 0.754 ** | 0.619 | 1 | | | | | | | | |
| K | −0.602 | 0.522 | −0.139 | 0.776 *** | 0.924 *** | 0.769 ** | 1 | | | | | | | |
| Mg | −0.444 | 0.645 * | −0.406 | 0.380 | 0.738 ** | 0.709 ** | 0.804 ** | 1 | | | | | | |
| P | −0.167 | −0.010 | −0.011 | 0.163 | 0.463 | 0.075 | 0.550 | 0.442 | 1 | | | | | |
| N | −0.741 ** | 0.571 | −0.104 | 0.640 * | 0.968 *** | 0.602 | 0.906 *** | 0.699 * | 0.410 | 1 | | | | |
| Fe | −0.428 | 0.259 | −0.189 | 0.525 | 0.859 *** | 0.506 | 0.904 *** | 0.725 ** | 0.813 ** | 0.827 *** | 1 | | | |
| Cu | 0.684 * | −0.536 | 0.009 | −0.625 * | −0.800 ** | −0.580 | −0.925 *** | −0.746 ** | −0.638 * | −0.836 *** | −0.887 *** | 1 | | |
| Mn | −0.741 ** | 0.486 | 0.000 | 0.753 ** | 0.906 *** | 0.497 | 0.911 *** | 0.570 | 0.562 | 0.915 *** | 0.858 *** | −0.900 *** | 1 | |
| Zn | −0.643 * | 0.350 | 0.302 | 0.797 ** | 0.768 ** | 0.565 | 0.866 *** | 0.447 | 0.578 | 0.811 ** | 0.808 ** | −0.854 *** | 0.918 *** | 1 |

**Table 4.** *Cont.*

| | ρb | AWC | pH | EC | SOC | Na | K | Mg | P | N | Fe | Cu | Mn | Zn |
|---|---|---|---|---|---|---|---|---|---|---|---|---|---|---|
| | | | | | Cover of *Vicia faba* | | | | | | | | | |
| ρb | 1 | | | | | | | | | | | | | |
| AWC | −0.599 ** | 1 | | | | | | | | | | | | |
| pH | 0.325 | −0.687 ** | 1 | | | | | | | | | | | |
| EC | −0.708 ** | 0.504 * | −0.490 | 1 | | | | | | | | | | |
| SOC | −0.829 *** | 0.729 ** | −0.641 ** | 0.840 *** | 1 | | | | | | | | | |
| Na | −0.122 | 0.602 ** | −0.482 | −0.026 | 0.297 | 1 | | | | | | | | |
| K | −0.738 *** | 0.862 *** | −0.668 ** | 0.802 *** | 0.847 *** | 0.335 | 1 | | | | | | | |
| Mg | −0.560 * | 0.531 * | −0.503 * | 0.264 | 0.440 | 0.381 | 0.528 * | 1 | | | | | | |
| P | −0.630 ** | 0.671 ** | −0.740 *** | 0.878 *** | 0.896 *** | 0.198 | 0.854 *** | 0.332 | 1 | | | | | |
| N | −0.735 *** | 0.666 ** | −0.517 * | 0.823 *** | 0.777 *** | 0.077 | 0.757 *** | 0.180 | 0.759 *** | 1 | | | | |
| Fe | −0.112 | 0.116 | −0.042 | 0.312 | 0.371 | −0.178 | 0.156 | −0.565 * | 0.380 | 0.349 | 1 | | | |
| Cu | −0.229 | 0.123 | 0.005 | 0.347 | 0.431 | −0.253 | 0.135 | −0.392 | 0.421 | 0.340 | 0.901 *** | 1 | | |
| Mn | −0.690 ** | 0.580 ** | −0.545 * | 0.824 *** | 0.867 *** | 0.074 | 0.708 ** | 0.090 | 0.822 *** | 0.840 *** | 0.655 ** | 0.631 ** | 1 | |
| Zn | −0.419 | 0.522 * | −0.243 | 0.525 * | 0.662 ** | 0.163 | 0.492 | −0.281 | 0.586 ** | 0.658 ** | 0.850 *** | 0.773 *** | 0.789 *** | 1 |
| | | | | | Cover of *Vicia sativa* | | | | | | | | | |
| ρb | 1 | | | | | | | | | | | | | |
| AWC | −0.563 * | 1 | | | | | | | | | | | | |
| pH | −0.096 | −0.534 * | 1 | | | | | | | | | | | |
| EC | −0.675 ** | 0.690 ** | −0.400 | 1 | | | | | | | | | | |
| SOC | −0.370 | 0.686 ** | −0.715 ** | 0.821 ** | 1 | | | | | | | | | |
| Na | 0.325 | −0.022 | 0.231 | −0.134 | −0.230 | 1 | | | | | | | | |
| K | −0.516 * | 0.714 ** | −0.587 * | 0.892 ** | 0.951 ** | −0.226 | 1 | | | | | | | |
| Mg | −0.670 ** | 0.445 | −0.128 | 0.803 ** | 0.677 ** | −0.090 | 0.747 ** | 1 | | | | | | |
| P | −0.366 | 0.267 | −0.186 | 0.760 ** | 0.616 * | 0.104 | 0.660 ** | 0.871 ** | 1 | | | | | |
| N | −0.580 * | 0.720 ** | −0.559 * | 0.819 ** | 0.849 ** | −0.279 | 0.908 ** | 0.692 ** | 0.539 * | 1 | | | | |
| Fe | −0.397 | 0.504 * | −0.680 ** | 0.596 * | 0.810 ** | −0.432 | 0.720 ** | 0.538 * | 0.395 | 0.709 ** | 1 | | | |
| Cu | −0.036 | 0.486 | −0.772 ** | 0.462 | 0.709 ** | −0.064 | 0.577 * | 0.159 | 0.209 | 0.448 | 0.622 * | 1 | | |
| Mn | −0.583 * | 0.731 ** | −0.332 | 0.880 ** | 0.719 ** | 0.002 | 0.810 ** | 0.811 ** | 0.754 ** | 0.767 ** | 0.454 | 0.177 | 1 | |
| Zn | −0.308 | 0.483 | −0.601 * | 0.775 ** | 0.921 ** | −0.219 | 0.860 ** | 0.781 ** | 0.766 ** | 0.724 ** | 0.745 ** | 0.533 * | 0.726 ** | 1 |

**Table 4.** *Cont.*

| | ρb | AWC | pH | EC | SOC | Na | K | Mg | P | N | Fe | Cu | Mn | Zn |
|---|---|---|---|---|---|---|---|---|---|---|---|---|---|---|
| | | | | | Cover of *Vicia sativa* and *Vicia ervilia* | | | | | | | | | |
| ρb | 1 | | | | | | | | | | | | | |
| AWC | −0.669 * | 1 | | | | | | | | | | | | |
| pH | 0.347 | −0.360 | 1 | | | | | | | | | | | |
| EC | −0.666 * | 0.619 | −0.349 | 1 | | | | | | | | | | |
| SOC | −0.768 ** | 0.767 ** | −0.394 | 0.802 ** | 1 | | | | | | | | | |
| Na | −0.405 | 0.610 | 0.220 | 0.184 | 0.447 | 1 | | | | | | | | |
| K | −0.614 | 0.779 ** | −0.335 | 0.849 *** | 0.767 ** | 0.527 | 1 | | | | | | | |
| Mg | −0.555 | 0.675 * | 0.058 | 0.233 | 0.601 | 0.892 *** | 0.582 | 1 | | | | | | |
| P | −0.790 ** | 0.904 *** | −0.553 | 0.774 ** | 0.904 *** | 0.467 | 0.875 *** | 0.642 * | 1 | | | | | |
| N | −0.840 *** | 0.656 * | −0.395 | 0.725 ** | 0.920 *** | 0.552 | 0.732 ** | 0.641 * | 0.831 ** | 1 | | | | |
| Fe | −0.678 * | 0.584 | −0.632 * | 0.797 ** | 0.828 ** | 0.069 | 0.561 | 0.124 | 0.731 ** | 0.781 ** | 1 | | | |
| Cu | −0.822 ** | 0.768 ** | −0.365 | 0.865 *** | 0.942 *** | 0.525 | 0.905 *** | 0.653 * | 0.928 *** | 0.932 *** | 0.738 ** | 1 | | |
| Mn | −0.750 ** | 0.652 * | −0.596 | 0.890 *** | 0.681 * | 0.033 | 0.725 ** | 0.105 | 0.777 ** | 0.638 * | 0.830 ** | 0.746 ** | 1 | |
| Zn | −0.895 *** | 0.661 * | −0.203 | 0.834 *** | 0.914 *** | 0.457 | 0.756 ** | 0.596 | 0.821 ** | 0.912 *** | 0.727 ** | 0.945 *** | 0.716 ** | 1 |

ρb, bulk density; AWC, available water content; EC, electrical conductivity; SOC, soil organic carbon. *, **, and *** significant at $p < 0.1$, $p < 0.05$, and $p < 0.01$, respectively.

Significant correlations ($p \leq 0.01$, $p \leq 0.05$, and $p \leq 0.1$) were found among many of the soil properties. The ρb was negatively correlated with all the soil properties, which coincides with Chaudhari et al. [128]. Similarly, the SOC had significant positive correlation with most of the soil physical and chemical properties studied, except bulk density, with a negative relationship (Table 4). This significant positive correlation among SOC and other soil properties implies that the increase of organic matter in soil under conservation agriculture practices improved soil physico-chemical properties. Finally, most macro and micronutrient contents were significantly positively correlated with each other, which is in line with findings reported by Jat et al. [122]. In general, all macro and micronutrients showed negative relationships with soil pH, as was stated by Kumar et al. [129].

3.2.2. Soil Microbial Activity

Figure 6 shows the soil microbial activity measured through fluorescein diacetate hydrolase (FDA) methodology. The highest enzymatic activity for both soil depths was found in plots with VS and VS-VE strategies. That is, VS enhanced their microbial activity compared to TT by 39% at 0–10 cm and 35% at 10–25 cm, while for VS-VE plots in both soil depths, this increase was 18 and 23%, respectively. On the contrary, VF presented a lower activity than TT. This may be due to the fact that the coverage achieved by this cover was not sufficient to improve the soil health. At both depths, the microbial activity with FDA was greater in NT than in TT, although without significant differences (at a confidence level of 95%). These findings are in concordance with the conclusions from a meta-analysis reported by Zuber and Villamil [130], who highlighted that no tillage and reduced tillage promote soil enzymatic activity. In this sense, Habig and Swanepoel [131] also concluded that the cropping system and the degree of soil disturbance significantly influence the enzymatic functions of the soil. Consequently, soil management strategies with less frequent soil disturbances imply greater soil organic matter, better hydric and thermal conditions, and a greater diversity of substrates, which favors microorganisms, as was found in the present experiment. In this context, Bhattacharyya et al. [132] stated that the accumulation of organic matter as a result of implementation of conservation

agriculture practices fostered soil microbial activity and singularly at upper soil layers. In a study by Roldán et al. [102], it was found that conservation agriculture techniques based on zero tillage and legume cover remarkably enhanced the soil enzymatic activity. In a cereal-based intensive cropping system, microbial activity FDA values under permanent beds and zero tillage were higher (20.1 and 21.7%, respectively) compared with conventional tillage [133]. In addition, Sofo et al. [134] showed how soil micro-organisms respond positively to sustainable management in a Mediterranean olive orchard.

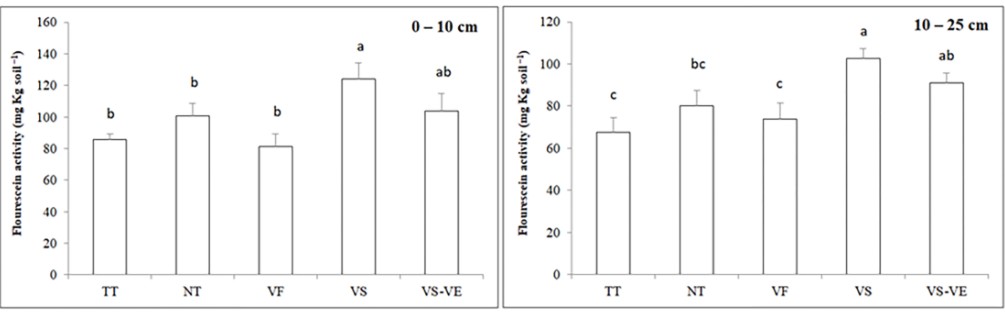

**Figure 6.** Average soil microbial activity in relation to soil management strategies applied in almond orchards. TT, traditional tillage; NT, no tillage; VF, cover of *V. faba*; VS, cover of *V. sativa*; VS-VE, cover of *V. sativa* and *V. ervilia*. Bars are standard deviation. Different letters indicate significant differences between management strategies by Tukey's test, $p < 0.05$.

*3.3. Foliar Analysis*

Assessing the impact of different soil management strategies on almond yield is quite complex due to the great interannual variability in crop yield owing to climatic conditions (annual rainfall, frost, hail, etc.). This causes the nutrient content in the leaves to be used as an indicator of crop yield. Table 5 shows the nutrient content from the foliar analysis. There were significant differences among the different SMSs for all the nutrients studied except for Cu, unlike the results obtained by Luján-Soto et al. [96], who only registered significant differences between regenerative agriculture and conventional tillage for leaf K content, with no significant differences for leaf N and P content.

**Table 5.** Macro and micronutrient contents in almond leaf by different soil management strategies.

| SMS | N | P | K | Zn | Fe | Mn | Cu |
|---|---|---|---|---|---|---|---|
| | (%) | | | (mg kg$^{-1}$) | | | |
| TT | 1.69 ± 0.07 bc | 0.11 ± 0.00 b | 1.30 ± 0.13 a | 21.80 ± 2.02 ab | 209.03 ± 15.87 c | 17.70 ± 1.45 bc | 5.66 ± 0.80 a |
| NT | 1.58 ± 0.11 c | 0.10 ± 0.00 b | 1.22 ± 0.08 a | 17.31 ± 1.20 c | 308.35 ± 34.85 a | 12.43 ± 3.03 c | 6.12 ± 0.76 a |
| VF | 1.86 ± 0.02 ab | 0.10 ± 0.01 b | 1.15 ± 0.06 ab | 20.08 ± 0.87 bc | 256.48 ± 31.67 abc | 11.72 ± 2.13 c | 6.03 ± 1.30 a |
| VS | 1.99 ± 0.23 a | 0.11 ± 0.00 b | 1.24 ± 0.22 a | 19.71 ± 1.43 bc | 228.69 ± 4.36 bc | 26.50 ± 4.36 a | 5.86 ± 0.23 a |
| VS-VE | 2.02 ± 0.09 a | 0.13 ± 0.01 a | 0.93 ± 0.07 b | 23.83 ± 1.13 a | 275.76 ± 22.83 ab | 21.88 ± 4.06 ab | 6.87 ± 0.76 a |

TT, traditional tillage; NT, no tillage; VF, cover of *V. faba*; VS, cover of *V. sativa*; VS-VE, cover of *V. sativa* and *V. ervilia*. Mean ± standard deviation. Different letters indicate significant differences between management strategies (Tukey's test, $p < 0.05$).

The leaf contents of N, P, and K obtained in the foliar analysis were within the values established as normal by Ferrández-Cámara et al. [135] according to the normal range for the mineral diagnosis of almond trees in Spain. The increase in leaf N content in treatments with plant cover coincided with the results of Sulas et al. [136], who registered an increase in the N content in the leaves in vineyards with legume cover. As in the study by De Leijster et al. [47] the N content in the leaves was lower for NT than TT. This is presumably because the NT increases soil compaction, which reduces the rate of N mineralization and crop root development, reducing plant nutrient uptake [137]. This could explain the lower N content for NT in both soil and leaves registered in this experiment.

The correlation between soil nutrients and leaf nutrient content was analyzed, and no significant relationship was found. This coincides with the results of Herencia and Maqueda [138], who found no correlation between soil and plant parameters in a study to determine the effect of time and the dose of organic fertilization on soil fertility.

### 3.4. Almond Yield and Quality

The crop average annual yield for each of the SMSs studied is shown in Figure 7. The almond yield during the study period was low and highly variable, ranging between ~30 and 550 kg ha$^{-1}$, depending on the SMS and the season. However, the great interannual variability in almond yield was fundamentally due to differences in annual precipitation, registering the highest production for all SMSs in 2018, which is the year that recorded the highest rainfall compared to the other four years with below-average precipitation. The differences in yield between the years 2019 and 2021 with the year 2020, despite having a very similar rainfall, may be due to alternate bearing behavior of almond trees, as has been concluded in other studies of rainfed almond in the Mediterranean region [139,140]. The low production recorded could be ascribed to the low rainfall recorded; likewise, during this season the plantation was affected by an aphid plague, cv. Marcona being highly susceptible to attack by this type of pest [141]. In addition, the treatments that can be used against these pests in organic farming are less effective than conventional chemical treatments, which can condition productivity.

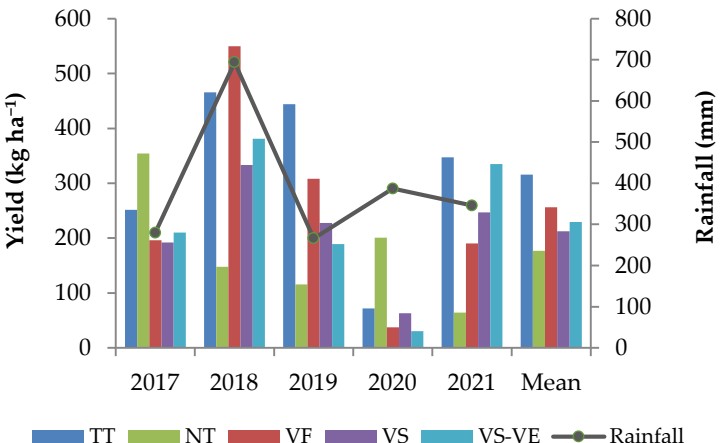

**Figure 7.** Annual almond yield response to different soil management strategies and rainfall depths for each growing season. TT, traditional tillage; NT, no tillage; VF, cover of *V. faba*; VS, cover of *V. sativa*; VS-VE, cover of *V. sativa* and *V. ervilia*.

The maximum average yield during the study period was recorded for TT with ~316 kg ha$^{-1}$, with reductions in the rest of the SMSs of 44.1, 32.8, 27.5, and 18.9% for NT, VS, VS-VE, and VF, respectively. In general, the yield was in line with those reported by Arquero [32] for the marginal rainfed almond orchards in the study area.

According to De Leijster et al. [47] and Martínez-Mena et al. [137], soil conservation practices may promote a lower crop yield in the first years after their implementation. That is, the possible negative effects of these practices on crop yields are related to the specific climatic conditions of the area, being more frequent in arid and semi-arid regions [21]. In general, the almond yield from different SMSs was in line with the AWC recorded during the monitoring seasons. In addition, factors such as the lower plant nutrient availability or high soil compaction might be ascribed the reduction in almond yield [137,142]. Therefore, it should be noted that obtaining reliable data of the impact of SMSs on rainfed almond yield requires long-term studies, given the large interannual variation of climatic conditions and the slow response of the soil to achieve any positive changes in quality and health terms [96].

The kernel yield, which can be defined as the percentage of seed with respect to the weight of the entire nut fruit, oscillated between 21.81 and 25.26% for VS and VF, respectively, although there are no significant differences among SMSs. Regarding the seasons, kernel yield oscillated between 26.33% and 22.62%, with significant differences between 2018 and the rest of the years. In this year was registered the highest kernel yield, coinciding with the highest amount of annual rainfall. The lower percentages obtained in the years with less precipitation coincide with the results of a study in Morocco by Melhaoui et al. [143] in which it was shown how the kernel yield of the cv. Marcona was strongly affected by drought. In general, the yields obtained in the present experiment was within the average range for hard shell varieties (<25%) established by Roncero et al. [144]. In this line, Muncharaz [145] stated that the shelling percentage for the cv. Marcona ranged between 22 and 28%, coinciding with values found in the present experiment.

Almond Quality Parameters

The effect of soil management strategies on moisture content, and the water activity of almonds are shown in Table 6.

**Table 6.** Average moisture content and water activity in almonds from each soil management strategy.

| SMS | Moisture | Water Activity |
|---|---|---|
|  | (%) | ($a_w$) |
| TT | 3.35 ± 0.20 b | 0.55 ± 0.05 n.s. |
| NT | 3.19 ± 0.08 b | 0.53 ± 0.05 n.s. |
| VF | 3.82 ± 0.26 a | 0.57 ± 0.06 n.s. |
| VS | 3.21 ± 0.23 b | 0.57 ± 0.05 n.s. |
| VS-VE | 3.29 ± 0.38 b | 0.52 ± 0.05 n.s. |

SMS, soil management strategy; TT, traditional tillage; NT, no tillage; VF, cover of *V. faba*; VS, cover of *V. sativa*; VS-VE, cover of *V. sativa* and *V. ervilia*. Values (mean of 3 replications) ± standard deviation; different letter within the same column are significant differences among SMSs; n.s., no significant differences by Tukey's test, $p < 0.05$.

There were no significant differences among SMSs for water activity ($a_w$). By contrast, significant differences between VF with the rest of the SMSs for the percentage of moisture was determined. The moisture content and $a_w$ values obtained were within the optimal industry standards for raw almonds, ranging between 3–6% and 0.3–0.6%, respectively [146,147]. Table 7 presents the results of texture and color analysis, showing significant differences for SMSs in all the determined parameters, except for the number of fractures. The cv. Marcona is hard-shelled as is preferred in the Mediterranean region, being more adapted to non-irrigated conditions, more resistant to predators and insects, and able to be stored for a long period without problems of excessive desiccation or rancidity [148,149].

Regarding the content of organic acids, significant differences ($p < 0.05$) for almonds from VF and VS-VE plots were found (Table 8). The content of organic acids obtained was higher than the content for cv. Vairo in a study on deficit irrigation in southwest Spain [150]. The total sugar content for almonds was not affected by SMSs, but the content of specific sugars was (Table 8). That is, the total sugar content ranged from 47.33 for NT to 52.88 g kg$^{-1}$ for the VS-VE plot, values that fall within the range established by a review of a world collection of almond samples (18–76 g kg$^{-1}$ almond) [51]. In addition, the prevalence of sucrose as the main sugar in almonds is in agreement with other studies [151–153].

The results of antioxidant activity (AA), total phenolic content (TPC), and fatty acids are shown in Table 9. The AA was studied through three methodologies, namely, ABTS, DPPH, and FRAP, since the use of several methods allows for a more reliable analysis of the antioxidant properties of the almond [154]. Only ABTS showed significant differences among TT and the rest of the SMSs. In addition, for the three methodologies, almonds from VS plots registered the highest value for AA and the lowest for TT plots. According to Bolling [52], each cultivar has its own antioxidant characteristics. However, the values

obtained in this study were higher than those obtained by Summo et al. [155], who obtained a higher AA by the DPPH method for cv. Marcona of 12.69 µmol Trolox $g^{-1}$.

Relative to the total phenolic content, there were significant differences among SMSs (Table 9). The highest TPC was recorded for VS-VE plots with 5.14 g GAE $kg^{-1}$, being 58, 47, 48, and 25% lower for TT, NT, VF, and VS, respectively. The values of the total phenolic content and the radical scavenging activity were much higher than those obtained by Čolić et al. [156] for cv. Marcona in a non-irrigated study in Serbia, with 0.20 mg GAE $g^{-1}$ and 0.81 mmol TE $kg^{-1}$ (DPPH), respectively. This may be due to the fact that the water stress to which the almond trees were subjected in our study due to the edaphoclimatic characteristics was greater, which increased the antioxidant activity [150]. In short, it is also difficult to compare the values of polyphenol content and antioxidant activity that can be found in the scientific literature, since they largely depend on the type of extraction solvent and standards used [157].

Fatty acids are divided into saturated fatty acids (SFA) and unsaturated fatty acids, which are subdivided into monounsaturated fatty acids (MUFA) and polyunsaturated fatty acids (PUFA) (Table 9). There are significant differences among SMSs both in total fatty acids and in their compositions. By comparing the findings with those obtained by Čolić et al. [158] in drought conditions, it is noted that the percentage of MUFA obtained in our study was much lower (57.5 compared to 74.8%) and instead increased the percentages of SFA and PUFA (16.8 compared to 9.31, and 25.7 compared to 15.9, respectively). Similar results were obtained by Kodad and Socias I Company [159] for cv. Marcona with MUFA, 72%; PUFA, 19.4%; and SFA, 8.6%, since the almond trees in rainfed conditions are continually subjected to important water stress. In this sense, various studies have shown how water stress is positively correlated with SFA and PUFA and negatively correlated with MUFA [150,153].

Table 10 shows the correlation coefficients among the main soil properties and the most important quality parameters of the almonds. Some parameters such as oxalic acid, glucose, or SFA were not significantly correlated with the soil properties analyzed. By contrast, a significant correlation was observed among SOC and the total content of sugars and TPC of almonds, while with the total organic acids and PUFA were negatively correlated. A positive relationship for soil N content and almond sugar was found, contrasting with negative correlations with total organic acids and PUFA. Similarly, extractable P showed a positive correlation with almond sugar content and TPC and was inversely correlated with organic tartaric and malic acid contents. The assimilable soil K, unlike P, was positively correlated with the content of tartaric acid; in addition, it was negatively correlated with the total content of polyphenols. Finally, the AWC showed a positive relationship with the sugar content, in contrast with the organic acid and unsaturated fatty acid contents.

In a study by Pérez-Murcia et al. [160], the agronomic effects of the application of composts in almond trees (cvs. Guara and Ferraduel) were analyzed, and it was concluded that these treatments enhanced the soil N, P, and K contents and produced almonds with higher sugars, organic acids, and fibers. This agrees with our findings for the sugar content, which increased with the SOC, N, and P contents. It did not coincide, however, with the contents of organic acids, which had significant negative relationships with the contents of SOC, N, and P. Only the soil K content was positively related to almonds citric and tartaric acid contents.

The total content of organic acids showed a significant negative correlation with AWC, which is in line with results obtained by Lipan et al. [84,161], with a positive correlation between the content of organic acids and water stress, although other authors concluded that there is no relationship between these parameters [152,162]. Total sugar content was positively correlated to AWC, which contradicts the results of various studies that concluded that there was a positive correlation between sugar content and water stress [84,161,163], with higher sugar content in deficit irrigation than in fully irrigated plots. These findings obtained in the present experiment conform to those studies with a higher content of sugars in irrigated almonds than in rainfed [162,164]. By contrast Nanos et al.

[153] concluded that there were no significant differences between rainfed and irrigated almond crops in terms of sugar content. Comparably, the results obtained by Lipan et al. [150] did not find significant relationships among antioxidant activity, TPC, and water availability in contrast with studies that reported an increase in antioxidant activity under water stress conditions in many crops, for example, in almonds [163], olives [165], or pistachios [166]. Moreover, numerous studies concluded that water stress has a significant influence on the composition of fatty acids [51,84]. In this study, taking into consideration that rainfed almond trees are in continuous water stress, a significant negative relationship between AWC and total fatty acids and unsaturated fatty acids (MUFA and PUFA) was found, there being no correlation with SFA. These results contradict others previously reported where irrigation increases the content of MUFA (oleic acid) [84,162]. Ours results agree with the results previously reported by Lipan et al. [161] and Gutiérrez-Gordillo et al. [163] with an increase in MUFAs and PUFAs with increasing water stress.

More medium and long-term studies for evaluating the impact of soil management strategies on yield and almond quality in both the current and subsequent seasons are needed. Since the influence of the harvest year on almond quality parameters has already been demonstrated [164], it will be essential to analyze the almond production in different years to dismiss the influence of this factor on the differences found for almond quality under the soil management strategies applied.

**Table 7.** Impact of different soil management strategies on texture and color parameters of almonds cv. Marcona.

| SMS | Kernel Cutting Force | | | | | Kernel Color Coordinates | | | | |
|---|---|---|---|---|---|---|---|---|---|---|
| | Fracturability (mm) | Hardness (N) | Work to Shear (Ns) | Average Force (N) | Number of Fractures | L* | a* | b* | C | Hue |
| TT | 1.65 ± 0.28 ab | 68.76 ± 15.59 ab | 56.40 ± 19.36 ab | 33.73 ± 8.43 ab | 12.48 ± 5.27 a | 45.63 ± 2.98 abc | 16.43 ± 0.89 a | 29.88 ± 2.68 ab | 34.13 ± 2.51 ab | 61.08 ± 2.23 bc |
| NT | 1.55 ± 0.33 abc | 67.98 ± 17.13 ab | 49.92 ± 17.55 bc | 32.44 ± 9.11 ab | 11.88 ± 5.17 a | 43.70 ± 3.00 c | 15.10 ± 1.78 b | 26.60 ± 3.73 c | 30.61 ± 3.97 c | 60.28 ± 2.23 c |
| VF | 1.71 ± 0.39 a | 73.11 ± 14.78 a | 64.33 ± 23.90 a | 36.48 ± 8.45 a | 13.50 ± 6.15 a | 44.22 ± 3.25 bc | 16.01 ± 1.14 a | 28.35 ± 2.67 bc | 32.57 ± 2.70 bc | 60.47 ± 1.86 c |
| VS | 1.49 ± 0.32 bc | 58.25 ± 9.75 c | 38.55 ± 10.85 c | 26.18 ± 5.86 c | 11.50 ± 5.15 a | 45.99 ± 4.43 ab | 16.48 ± 1.93 a | 31.30 ± 4.49 a | 35.39 ± 4.78 a | 62.11 ± 1.65 ab |
| VS-VE | 1.37 ± 0.34 c | 60.57 ± 14.66 bc | 40.23 ± 14.88 c | 28.91 ± 6.96 bc | 11.78 ± 5.89 a | 46.56 ± 4.82 a | 16.16 ± 1.64 a | 31.47 ± 3.22 a | 35.40 ± 3.43 a | 62.78 ± 1.81 a |

SMS, soil management strategy; TT, traditional tillage; NT, no tillage; VF, cover of *V. faba*; VS, cover of *V. sativa*; VS-VE, cover of *V. sativa* and *V. ervilia*. Values (mean of 20 replications) ± standard deviation; different letters within the same column are significant differences among SMSs by Tukey's test, $p < 0.05$. L*, a*, b* are color coordinates; C, chroma.

**Table 8.** Impact of different soil management strategies on organic acids and sugars content of almonds cv. Marcona.

| SMS | Organic Acids (OA) | | | | | Sugars | | | |
|---|---|---|---|---|---|---|---|---|---|
| | Oxalic | Citric | Tartaric | Malic | ΣOA | Sucrose | Glucose | Fructose | ΣSugars |
| | (g kg⁻¹) | | | | | (g kg⁻¹) | | | |
| TT | 2.03 ± 0.09 a | 3.10 ± 0.14 abc | 2.48 ± 0.10 a | 6.33 ± 4.23 ab | 13.95 ± 4.39 ab | 36.23 ± 2.62 a | 6.64 ± 1.50 b | 6.52 ± 1.11 ab | 49.38 ± 2.18 a |
| NT | 1.90 ± 0.07 a | 2.66 ± 0.39 c | 2.14 ± 0.15 ab | 7.17 ± 0.84 ab | 13.88 ± 1.34 ab | 27.84 ± 2.37 b | 14.18 ± 0.40 a | 6.50 ± 0.77 ab | 48.50 ± 3.28 a |
| VF | 2.09 ± 0.11 a | 3.56 ± 0.16 a | 2.32 ± 0.13 ab | 9.45 ± 4.32 a | 17.41 ± 4.45 a | 26.34 ± 1.68 b | 15.32 ± 1.24 a | 5.68 ± 0.68 b | 47.33 ± 2.99 a |
| VS | 2.04 ± 0.12 a | 3.39 ± 0.58 ab | 2.01 ± 0.26 b | 3.85 ± 0.42 ab | 11.29 ± 1.25 ab | 29.58 ± 2.55 b | 14.94 ± 1.14 a | 6.96 ± 1.06 ab | 51.45 ± 4.59 a |
| VS-VE | 1.98 ± 0.04 a | 2.78 ± 0.06 bc | 1.56 ± 0.22 c | 3.41 ± 0.26 b | 9.73 ± 0.44 b | 37.98 ± 2.26 a | 6.86 ± 1.02 b | 7.74 ± 1.04 a | 52.58 ± 2.27 a |

SMS, soil management strategy; TT, traditional tillage; NT, no tillage; VF, cover of *V. faba*; VS, cover of *V. sativa*; VS-VE, cover of *V. sativa* and *V. ervilia*. Values (mean of 20 replications) ± standard deviation; different letter within the same column are significant differences among SMSs by Tukey's test, $p < 0.05$.

**Table 9.** Impact of soil management strategies on the antioxidant activity, total phenolic content, and fatty acids of almonds cv. Marcona.

| SMS | ABTS$^{\bullet+}$ | DPPH$^{\bullet}$ | FRAP | TPC | SFA | MUFA | PUFA | Total Fatty Acids |
|---|---|---|---|---|---|---|---|---|
| | (mmol Trolox kg$^{-1}$) | | | (g GAE kg$^{-1}$) | (g kg$^{-1}$) | | | |
| TT | 7.27 ± 0.67 b | 13.99 ± 1.27 a | 7.71 ± 1.66 a | 2.16 ± 1.23 c | 39.36 ± 2.01 a | 144.56 ± 10.09 a | 65.13 ± 4.52 a | 249.25 ± 15.88 a |
| NT | 10.42 ± 3.00 a | 14.84 ± 0.75 a | 8.34 ± 2.14 a | 2.73 ± 1.25 bc | 33.78 ± 4.59 b | 121.08 ± 11.69 bc | 58.03 ± 6.30 bc | 213.00 ± 22.58 b |
| VF | 11.08 ± 1.63 a | 16.03 ± 2.98 a | 8.97 ± 1.43 a | 2.65 ± 0.49 bc | 36.89 ± 1.15 ab | 132.87 ± 8.36 ab | 63.28 ± 2.92 ab | 233.17 ± 7.57 a |
| VS | 12.35 ± 1.18 a | 16.11 ± 0.63 a | 10.50 ± 2.56 a | 3.88 ± 1.61 ab | 35.22 ± 1.38 b | 114.12 ± 14.10 c | 52.20 ± 6.56 c | 201.50 ± 12.79 b |
| VS-VE | 11.22 ± 3.06 a | 14.68 ± 0.33 a | 10.44 ± 2.02 a | 5.14 ± 1.03 a | 38.89 ± 1.81 a | 139.01 ± 3.68 a | 61.07 ± 1.93 ab | 238.75 ± 4.50 a |

SMS, soil management strategy; TT, traditional tillage; NT, no tillage; VF, cover of *V. faba*; VS, cover of *V. sativa*; VS-VE, cover of *V. sativa* and *V. ervilia*; ABTS$^{\bullet+}$, 2.2-azino-bis (3-ethylbenzothiazoline-6-sulfonic acid); DPPH$^{\bullet}$, 2,2-diphenil-1-picrylhydrazyl; FRAP, ferric reducing ability of plasma; TPC, total phenolic content; SFA, saturated fatty acids; MUFA, monounsaturated fatty acids; PUFA, polyunsaturated fatty acids. Values (mean of 3 replications) ± standard deviation; different letters within the same column are significant differences among SMSs by Tukey's test, $p < 0.05$.

**Table 10.** Relationships among selected soil properties and almond quality parameters.

| Soil Parameter | Almond Quality Parameters | | | | | | | | | | | | | | | | |
|---|---|---|---|---|---|---|---|---|---|---|---|---|---|---|---|---|---|
| | Oxalic | Citric | Tartaric | Malic | ΣOA | Sucrose | Glucose | Fructose | ΣSugars | ABTS$^{\bullet+}$ | DPPH$^{\bullet}$ | FRAP | TPC | SFA | MUFA | PUFA | Total Fatty Acids |
| SOC | −0.236 | −0.363 * | −0.723 *** | −0.658 *** | −0.700 *** | 0.555 *** | −0.388 | 0.623 *** | 0.565 *** | 0.216 | −0.086 | 0.383 * | 0.607 *** | 0.041 | −0.179 | −0.440 ** | −0.250 |
| N | −0.123 | −0.164 | −0.410 * | −0.613 *** | −0.607 *** | 0.445 ** | −0.281 | 0.501 ** | 0.495 ** | 0.115 | −0.054 | 0.292 | 0.392 * | 0.003 | −0.261 | −0.497 ** | −0.325 |
| P | 0.335 | 0.355 | −0.430 ** | −0.354 | −0.305 | 0.248 | −0.082 | 0.268 | 0.358 | 0.342 | 0.201 | 0.448 ** | 0.479 ** | 0.241 | −0.039 | −0.226 | −0.063 |
| K | 0.310 | 0.267 | 0.544 *** | 0.079 | 0.157 | 0.357 | −0.475 ** | −0.141 | −0.076 | −0.515 ** | −0.186 | −0.269 | −0.348 | 0.484 ** | 0.498 ** | 0.421* | 0.533 ** |
| AWC | −0.190 | −0.201 | −0.320 | −0.566 *** | −0.563 *** | 0.334 | −0.173 | 0.446 ** | 0.434 ** | 0.098 | −0.053 | 0.231 | 0.297 | −0.118 | −0.359 | −0.551 *** | −0.424 ** |

Correlation coefficient; *, **, and *** significant at $p < 0.1$, $p < 0.05$, and $p < 0.01$, respectively. SOC, soil organic carbon; AWC, available water content; ΣOA, total organic acids; ABTS$^{\bullet+}$, (2.2-azino-bis (3-ethylbenzothiazoline-6-sulfonic acid); DPPH$^{\bullet}$, (2,2-diphenil-1-picrylhydrazyl); FRAP, ferric reducing ability of plasma; TPC, total phenolic content; SFA, saturated fatty acids; MUFA, monounsaturated fatty acids; PUFA, polyunsaturated fatty acids.

## 4. Conclusions

The low productivity of rainfed agricultural systems is the main factor that accentuates their abandonment. Taking into consideration the high rainfall dependence, farmers cultivate their crops with high vulnerability to recurrent droughts. Facing this challenge in a context of a changing climate is urgent in terms of investing resources and efforts in sustainable development and adaptation measures for high-risk rainfed woody crops through harvesting runoff rainwater and boosting the soil health and quality.

The results obtained in the present experiment suggest that the additional implementation of conservation agriculture practices, such as legume cover crops, leads to a gradual enhancement of soil quality for organic rainfed almond orchards located in marginal areas. Specifically, an improvement has been observed in the physical (bulk density, AWC, and aggregate stability), chemical (soil-organic carbon, N, K, and micronutrients content), and biological (greater microbial activity) properties of the soil.

Regarding the influence of soil management strategies on almond quality parameters, no decisive results were obtained. This may be due to the fact that only the almonds from the last season were analyzed, suggesting that the use of legume cover improves the antioxidant activity of almonds and total polyphenols content.

Given the serious challenges that agriculture currently faces, it is necessary to apply soil management strategies that guarantee the long-term sustainability of agroecosystems, conserving and/or improving soil health. Thus, long-term studies are crucial to prove the great real potential for the improvement of soil quality based on the use of legume cover in rainfed almonds and allow inferences regarding their effects on the nutritional almond composition.

**Author Contributions:** Conceptualization, B.C.R., V.H.D.Z., J.F.H.G., M.S. and I.F.G.-T.; methodology, B.C.R., V.H.D.Z., J.F.H.G., M.S., L.L., F.H., E.S. and Á.A.C.-B.; investigation, B.C.R., V.H.D.Z., J.F.H.G., L.L., M.S., B.G.R. and I.F.G.-T.; resources, B.G.R.; validation, L.L., F.H., E.S. and Á.A.C.-B.; writing—original draft preparation, B.C.R., J.F.H.G., L.L. and I.F.G.-T.; writing—review and editing, B.C.R. and V.H.D.Z. All authors have read and agreed to the published version of the manuscript.

**Funding:** The first author, Belén Cárceles Rodríguez, has a contract co-financed by the National Institute for Agricultural and Food Research and Technology (FPI-INIA 2018) and the European Social Fund (ESF) "The European Social Fund invests in your future". This publication was sponsored by the following research project: "Integrated soil and water management in rainfed almond in a context of sustainable agriculture" (RTA2017-00097-00-00) granted by INIA (National Institute for Agricultural and Food Research and Technology) and MINECO (Spanish Ministry of Economy and Competitiveness).

**Data Availability Statement:** The analyzed datasets are available from the corresponding author on reasonable request.

**Conflicts of Interest:** The authors declare no conflict of interest.

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
