# Peer review of "Soil Management Strategies in Organic Almond Orchards: Implications for Soil Rehabilitation and Nut Quality"

_agronomy, doi:10.3390/agronomy13030749_

Round 1

Reviewer 1 Report

The paper presents an important overall theme; however, the experimental details are vague in parts. The attached document shows where improvements are required. 

Author Response

We thank the reviewer for the appreciation of our manuscript, and for his comments by it, as they will undoubtedly contribute to improving it.

Reviewer 2 Report

This study focused on analyzing the effects of different soil-management strategies on selected physical, chemical, and biological soil properties, relevant to soil health, and its implications on almond yield and quality. The topic of this research is interesting and of importance, as it provides field-based experiment results suggesting that the additional implementation of conservation agriculture practices such as legume cover crops, leads to a gradual enhancement of soil quality for organic rainfed almond orchard located in marginal areas. Authors did the great job for introduction and methodology. However, the problems of the manuscript are in the results section. Please see my comments below:

Line 65-67: The sentence doesn’t convey the same message that have in abstract of the reference 17. For example, it says that the 12 million hectares of agricultural areas in the EU that suffer from severe erosion. Soil erosion can be slight, moderate, or severe.

Line 67: 1.25 billion not million

Line 90-91: Reference is required.

Line 117: need comma after citation

Line 138-139: does not need “ through of different soil management strategies”.

Line 151: delete “de”

Section 2.2. Are these experiment plots created in 2016 or before that?

Line 166: … plots were …

Line 171: What tools were used to collect undisturbed soil cores? Also, need to specify model and company of the tools inside parenthesis.

Table 2. Caption could be “Soil-management strategies applied in almond organic production system during study period (2016-2021)” to make it clear.

Line 193: Is it 0.5? What is the purpose of analysis water content at 0.5 kPa pressure?

Line 219: were determined

Line 262: 15 g and test …..

Line 322-510: Section 3.1 to 3.2.1

I have major concern in these sections because findings are not properly described. It is bit confusing.

-What are main regions of showing 2016 and 2021 results separately for each SMS in these sections not for other sections?

-Objective was to assess the effects of SMS on soil properties. It would be better to show the effects of SMS by combining both year data. Were you investigating year and depth wise effect too?

Line 326: ….VS-VE, NT, and VF treatments registered a decrease in ρb of 8.1, 3.1, and 0.4%, respectively. This is bit confusing. For example, in this sentence, it said a decrease in bulk density of 8.1,….. So, compared to what? What is your reference point? Is it compared 10-25 cm depth or is compared 2021 values?

Line 336: lower sand content ….but compared to what?

In figure 2, unit is required to mention in the y-axis inside parenthesis. E.g. AWC (…). Also, need to have same y-axis range for both AWC graphs (e.g., 20-40 for 0-10 cm and 10-25 cm soil depths).

In the figure, you have two letters above error bar, but you haven’t explained about its meaning. What does that mean? E.g., if you have letters a, ab, and b, how do you explain? Is it significant difference or not?

Line 354: in figure 2 caption, you have mentioned during 5-year monitoring period, but in the graph you have two years (2016 and 2021) only. The caption need to be modified for clarification.

Figure 6: need y-axis title

Line 640: y?

Line 722: what context?

Line 723: it says “concluded that these organic treatments …”. What organic treatments? Is it mentioned earlier?

Line 733: … is no relationship ….

Line 797-793: In conclusion, I have not seen the summary of the results. It would be better to provide some main findings in the conclusion.

Author Response

(The authors gave the same response as above.)
